# Advances in Regeneration of Retinal Ganglion Cells and Optic Nerves

**DOI:** 10.3390/ijms22094616

**Published:** 2021-04-28

**Authors:** Fa Yuan, Mingwei Wang, Kangxin Jin, Mengqing Xiang

**Affiliations:** 1State Key Laboratory of Ophthalmology, Guangdong Provincial Key Laboratory of Ophthalmology and Visual Science, Zhongshan Ophthalmic Center, Sun Yat-sen University, Guangzhou 510060, China; yuanf28@mail2.sysu.edu.cn (F.Y.); wangmw7@mail2.sysu.edu.cn (M.W.); 2Guangdong Provincial Key Laboratory of Brain Function and Disease, Zhongshan School of Medicine, Sun Yat-sen University, Guangzhou 510080, China

**Keywords:** retinal ganglion cell, optic nerve, axon regeneration, ESC, iPSC, reprogramming

## Abstract

Glaucoma, the second leading cause of blindness worldwide, is an incurable neurodegenerative disorder due to the dysfunction of retinal ganglion cells (RGCs). RGCs function as the only output neurons conveying the detected light information from the retina to the brain, which is a bottleneck of vision formation. RGCs in mammals cannot regenerate if injured, and RGC subtypes differ dramatically in their ability to survive and regenerate after injury. Recently, novel RGC subtypes and markers have been uncovered in succession. Meanwhile, apart from great advances in RGC axon regeneration, some degree of experimental RGC regeneration has been achieved by the in vitro differentiation of embryonic stem cells and induced pluripotent stem cells or in vivo somatic cell reprogramming, which provides insights into the future therapy of myriad neurodegenerative disorders. Further approaches to the combination of different factors will be necessary to develop efficacious future therapeutic strategies to promote ultimate axon and RGC regeneration and functional vision recovery following injury.

## 1. Introduction

Glaucoma is the second leading cause of blindness worldwide, which affects about 79.6 million people by 2020 [1]. Glaucoma is an incurable neurodegenerative disorder marked by selective, progressive, and irreversible degeneration of retinal ganglion cells (RGCs) and the optic nerve [2]. RGCs function as the only output neurons conveying the detected light information from the retina along their axons within the optic nerve to the retinorecipient areas such as the superior colliculus and lateral geniculate nucleus, which is a bottleneck of vision formation. Novel RGC types were uncovered in succession. Over 40 subtypes of RGCs have been identified up to now based on their anatomical, physiological, and transcriptomic properties [3,4,5,6]. Each subtype encodes distinct and specific features of the visual scene in parallel for transmission to the central targets in the brain [3], and RGC subtypes differ dramatically in their ability to survive and regenerate following injury [5,7]. The death or dysfunction of RGCs causes irreversible blindness ultimately because surviving RGCs lose the intrinsic capacity to regenerate themselves and their axons, similar to most neurons in the mammalian mature central nervous system (CNS) [8].

Therefore, therapeutic strategies that support visual restoration have focused on protecting RGCs from degeneration, promoting RGC and axon regeneration after injury, and reestablishing their correct projection relationships. In the past few years, numerous studies have focused on promoting the regeneration of RGCs and their axons and have made great progress. Researchers have identified many factors that suppress or promote RGC survival and optic nerve regeneration. Much attention has been drawn to reprogramming since induced pluripotent stem cells (iPSCs) were created and used in regenerative medicine. The present review addresses recent advancements in this field of RGC and axon regeneration including (i) factors that affect axon regeneration; (ii) the differentiation of embryonic stem cells (ESCs) and iPSCs into cells with specific RGC characteristics; and (iii) the regeneration of RGCs by reprogramming.

## 2. Factors That Affect RGC Axon Regeneration

Axon regeneration after injury goes through the following processes: sensing injury stimulation, resealing the ruptured membrane, remodeling microtubules, rearrangements of cytoskeleton and neurofilaments, reorganization of organelles, the formation of growth cones, and extension of axons toward the targets [9]. Inflammation also plays a role in axon regeneration in some types of neurons. Cell-intrinsic and -extrinsic factors related to these processes affect axon regeneration to varying degrees. Overcoming the intrinsic suppressors, enhancing the intracellular axon regenerative signaling, and/or ameliorating the extracellular environments are prevalent manipulations used for stimulating RGC axon regeneration.

### 2.1. Microtubules, Cytoskeleton, and Neurofilaments

It is not surprising that the intervention of microtubules, cytoskeleton, and neurofilaments influence axon regrowth given the aforementioned processes. Providing taxol, an approved microtubule stabilization drug, can promote spinal cord axon regeneration and functional restoration [10]. Chen et al. identified exchange factor for Arf6 (EFA-6) as an intrinsic inhibitor of regrowth via microtubule dynamics [11]. Recently, Ruschel et al. showed that epothilone B, an approved drug that can cross the blood–brain barrier, promoted functional axon regeneration by inducing concerted microtubule polymerization [12].

### 2.2. Transcription Factors

Transcription factors (TFs) control neuronal growth, cell differentiation and survival, and synapse stabilization during development. In addition, they play a key role in axon regeneration. The axon regenerative ability is limited because of multitudinous changes in TF expression [13]. Prominent axon regeneration can be achieved by manipulating different TFs. Previous studies have shown that the Klf family can regulate intrinsic axon regeneration ability, and there are complicated interactions in the Klf family in this program. For instance, the overexpression of Krüppel-like factor 4 (Klf4) and Klf9 suppresses neurite growth, but the overexpression of Klf6 and Klf7 promotes neurite growth [14]. Likewise, Sox11 delivered by intravitreal injection of adeno-associated viruses (AAVs) to mouse retinas promotes axon regeneration of non-αRGCs but kills αRGCs [15]. By contrast, AAV-mediated osteopontin (OPN) plus insulin-like growth factor 1 (IGF1) promotes αRGC axon regeneration [7].

### 2.3. Signaling Pathway

By manipulating different signaling pathways, robust axon regrowth has been accomplished. In fact, as RGCs mature, mTOR is downregulated by the negative regulators SOCS3 and PTEN. Combining neural activity with activation of mTOR can enhance axon regrowth [16,17]. For instance, both deleting PTEN alone [18,19,20] and the co-deletion of PTEN and SOCS3 [21] triggered robust axon regeneration after injury. Forced expression of melanopsin in the RGCs induced axonal regeneration after optic nerve crush (ONC) by activation of the mTOR signaling pathway, which was comparable to PTEN knockdown. Silencing them with Kir2.1 remarkably curbed mTOR signaling and axon regeneration [17]. The combination of manipulating HDAC5 phosphorylation state to activate the mTOR pathway and the stimulation of the immune system promoted robust axon regeneration as well [22]. The enhancement of mTOR signaling by *Pten* deletion combined with cAMP and Zymosan treatments promoted the regeneration of long RGC axons capable of reaching the brain target areas [23]. By contrast, Sema3A greatly restrained axon regeneration via ROCK2 in RGCs, and this repression could be ameliorated by blocking ROCK2 [24]. Altered TGFβ signaling is also involved in axon regeneration and functional recovery [10]. In addition, the Wnt signaling pathway is critical in axon regeneration. Intravitreal injection of Wnt3a induced axon regeneration after optic nerve injury in vivo and reducing expression of Stat3 diminished Wnt3a-dependent axonal regeneration and RGC survival [25]. Furthermore, Park et al. found that the bulk of injured RGCs displayed activated Stat3 and survived with comparatively high spontaneous axon regeneration in an adult naked mole-rat ONC model [26]. Additionally, a few studies show that signaling mechanisms implicated in cancer may improve axonal regeneration in the optic nerve [27]. Armcx1 promoted both neuronal survival and axon regeneration after injury [28]. Inhibition of RelA either in astrocytes or oligodendrocytes (ODCs), or in neurons and macroglia together, elicited a graded cell-type-specific stimulation of axon regrowth [29].

### 2.4. Inflammatory Stimulation

Injury-derived inflammatory signals that recruit cellular and molecular processes have been shown to affect the intrinsic regenerative ability. The stimulation of inflammation-related cytokines such as interleukin 6 (IL-6) and ciliary neurotrophic factor (CNTF) can induce axon regeneration. More recently, Bei et al. showed that a combination of OPN, IGF1, and CNTF induced regrowth of retinal axons, which is similar to the effects of PTEN inhibition [30]. Increased CNTF levels in the retina brought about enhanced RGC axon regeneration [31]. Combining CNTF and RhoA shRNA could significantly stimulate axon regeneration, while only decreasing RhoA expression in the ONC model had impotent effects on axon regeneration [32]. Likewise, RGC axon regeneration could also be achieved by a cocktail treatment of CNTF and Y-27632, which is a ROCK-inhibitor [33]. In addition, AAV-mediated hIL-6 delivery enabled long-distance axon regeneration, with some axons growing through the optic chiasm [34]. Importantly, leucine-rich repeat and immunoglobulin-like domain-containing Nogo receptor-interacting protein 1 (LINGO-1) plays an important role in nerve fiber regeneration, and LINGO-1 deletion cooperates with the positive microenvironment in axon regeneration [35]. Both anti-LINGO-1 antibody therapy and the genetic deletion of LINGO-1 reduced nerve crush-induced axonal degeneration and enhanced axonal regeneration [36].

### 2.5. Exogenous Growth and Neurotrophic Factors

Growth and neurotrophic factors have different effects on axon regeneration. A recently identified unique granulocyte subtype, with characteristics of an immature neutrophil, can drive axon regeneration by the secretion of a series of growth factors [37]. Moreover, hepatocyte growth factor has been shown to promote neuronal survival and axonal regeneration [38]. Fibroblast growth factor has been reported to increase axonal sprouting [39]. In vivo, the calcium-binding protein oncomodulin, as a potent macrophage-derived growth factor for RGCs and other neurons, promotes regeneration in the mature rat optic nerve [40]. On the other hand, intraocular injection of BDNF or neurotrophins did not stimulate RGC survival or axon regeneration [41,42].

### 2.6. Epigenetic Modifications

Epigenetic modifications are involved in neuron and axon regeneration and different types of epigenetic modifications may contribute to the differentiation and regeneration of different cell lineages. This is not surprising because the regeneration program is regulated by a temporally changing cast of TFs that bind to stably accessible DNA regulatory regions [43]. Researchers showed that Ascl1a, Apobec2a, and Apobec2b contributed to optic nerve regeneration, which implies that DNA demethylation may underlie the regenerative program [44]. Indeed, the forced expression of Oct4, Sox2, and Klf4 in RGCs, which induce global DNA demethylation, can promote axon regeneration in the ONC model [45]. Meanwhile, the ectopic expression of Ascl1 in the Müller glia (MG) together with a histone deacetylase inhibitor enables mice to generate neurons after NMDA damage [46].

### 2.7. Extracellular Proteins

Multiple extracellular proteins, including myelin proteins, integrin, and kinases, create an intricate environment for axon regeneration. Various myelin proteins have an inhibitory effect on axon regeneration, including “Nogo” and semaphorins [47,48]. Although the neutralization of them could promote RGC axon regeneration in vitro [47], removing these proteins in vivo has little or no effect on regeneration [49]. However, neutralizing Nogo can enhance regeneration if RGCs are shifted into a growth state [50]. The regeneration of sensory axons in the spinal cord can be achieved by the expression of integrin together with kindlin-1, which alone has an activity to promote sensory axon regeneration [51]. The inhibition of matrix metalloproteinases (MMPs) significantly increased RGC axon regeneration. Focal adhesion kinase (FAK) inhibition reduced RGC survival and abrogated the neuroprotective effects of MMP inhibition, whereas FAK activation increased RGC survival despite MMP activation [52]. Therefore, the effects of FAK were dominant over those of MMPs. The overexpression of THBS1 in RGCs enhanced regeneration in both ipRGCs and non-ipRGCs [53].

### 2.8. Ions and Ion Channels

Mobile ions and ion channels also have a place in axon regeneration. The dysregulation of amacrine-derived Zn^2+^ signaling disrupts RGC regeneration [54,55]. The deletion of *slc30a3*, the gene encoding ZnT-3, and the intraocular injection of Zn^2+^ chelators allows considerable axon regeneration [55]. Furthermore, TPEN (N,N,N’,N’-tetrakis(2-pyridyl methyl) ethylenediamine), a Zn^2+^ chelator, in combination with Klf9 knockdown promotes RGC survival and axon regeneration [56]. Additionally, the application of calcium channel inhibitors after ONC preserved axonal integrity and promoted axon regeneration [57]. The deletion or silence of Cacna2d2, which encodes a subunit of voltage-gated calcium channels, promoted axon growth in vitro [58]. The pharmacological blockade of the voltage-gated calcium channel with Pregabalin enhanced axon regeneration after spinal cord injury [58].

### 2.9. Neural Activity

The activities of neurons have a positive effect on axon regrowth. Brief low-frequency electrical stimulation enhances nerve regeneration and targets reinnervation [59]. Increasing activity in ipRGCs has recently been shown to enhance RGC regrowth [17], and electrical stimulation has been shown to prolong both the survival and function of RGCs in various models of ophthalmic disease [60]. However, the efficiency of axon regeneration by enhancing neuronal activity alone is unsatisfactory. To improve the efficiency of regeneration and re-establish accurate circuit connections, a new strategy has been developed, which combines enhanced neural activity via visual stimulation with activation of mTOR signaling to allow for long-distance and target-specific regeneration [16]. Therefore, cocktail therapy is likely a trend in the future.

## 3. RGC Regeneration

In the past two decades, researchers have made great progress in the field of in vitro and in vivo RGC induction or regeneration. Different strategies (Figure 1) and a large number of protocols (Table 1) that support RGC regeneration have been proposed. In this section, we will focus on regeneration of RGCs via the differentiation of ESCs and iPSCs as well as direct reprogramming.

### 3.1. Generation of RGCs from Stem Cells

#### 3.1.1. ESCs

Mammals, unlike teleost fish, cannot regenerate the retina after various types of injury. Consequently, numerous in vitro protocols of RGC differentiation from pluripotent stem cells were created [83,84]. Although the supposedly mammalian adult retinal stem cells were found in the pigmented ciliary margin [85], there is no irrefutable evidence that they are able to regenerate RGCs damaged by injury or disease, which is in contrast to the situation in zebrafish and amphibians. In the chicken, Fischer and colleagues found that RGCs could be induced from retinal margin cells of post-hatch chicken by the co-injection of insulin and FGF2 into the vitreous chamber [61].

bFGF-induced ESCs were able to generate RGC-like cells upon differentiation, which were capable of integrating into the host retina [86]. In addition, hESCs can be directed to retinal progenitors by a combination of Noggin, Dkk1, and IGF1, which subsequently differentiated primarily into ganglion and amacrine cells [65]. Another protocol adapted from previous work [65] extended the culture duration of the embryoid body and used 10% knockout serum replacement, resulting in an enriched population of functional RGCs from hESCs [68]. Sluch et al. described a protocol that led to the differentiation of hESCs to RGCs and their subsequent isolation [67], benefiting from a modified photoreceptor differentiation protocol [87]. Together with a novel Brn3b-tdTomato-Thy1.2 reporter line, they designed an original protocol called DIDNF+D (Dorsomorphin + IDE2 + Nicotinamide + Forskolin + DAPT) that can improve the efficiency of the differentiation and purification of stem cell-derived RGCs [69]. 

#### 3.1.2. iPSCs

The establishment and development of iPSCs [88,89,90,91,92] and organoid culture systems [93,94,95,96,97], which mimic organogenesis in vitro, hold a great promise for a range of biological and biomedical applications, especially for regenerative medicine, by removing the limitation of replacement therapies and by enabling the development of in vitro disease models for drug screening. As an excellent cell resource, patient-derived iPSCs allow the regeneration of different and sufficient quantities of autologous cell types almost without the risk of immune rejection and iPSCs have already been generated from patients with multifarious diseases [98]. In addition, the procedure to generate urine-derived iPSCs with high reprogramming efficiency has been established [99], which provides a promising noninvasive source of stem cells and can subsequently differentiate into desired cell types.

By mimicking RGC genesis, Deng et al. performed a stepwise and efficient differentiation of human Tenon’s capsule fibroblasts-derived iPSCs toward RGC-like cells by combining DLN (Dkk1 + Lefty A + Noggin) treatment and Atoh7 overexpression sequentially [78]. iPSCs can differentiate into RGCs in neural induction and retinal differentiation culture medium [100]. The overexpression of a single gene can achieve the same effect. Mouse ESCs or iPSCs were able to be induced into RGCs by Pax6 overexpression and subsequent limiting-dilution culture [66,74]. In addition, retrovirus-mediated Neurod1 overexpression in iPSCs together with retinoic acid and taurine treatment increased the expression of RGC markers [75]. Similar to this, a single chemical, DAPT, can induce Pax6/Rx-positive stem cells to undergo differentiation into functional RGCs [72]. 

However, the potential risks of iPSCs such as genomic instability and immunogenicity differences [101,102,103,104] cannot be ignored. Whilst this might seem to be a truism, it is nevertheless a crucial problem to address. 

#### 3.1.3. Organoids

Since 3D retinal organoids were successfully established from mouse [96] and human ESCs [105], a number of adapted protocols were established for specific research purposes, such as generating retinal organoids from iPSCs [106,107], the formation of specific structures [107,108,109], or the formation of cell-specific features [87,108,109,110]. This suggests a possibility to restore vision via the transplantation of RGCs gained from retinal organoids because the efficient derivation of sufficient numbers of functional and integration-competent cells might partly remove a key limitation for regenerative medicine. Indeed, cells from ESC-derived eye-like structures were integrated into the RGC layer and differentiated into neurons when transplanted into adult eyes [73]. Moreover, Tanaka et al. efficiently generated self-induced RGCs with functional axons from mouse and human iPSCs by combining the cultivation of 3D floating aggregates with a subsequent 2D adhesion culture [70,71].

### 3.2. Generation of RGCs by Reprogramming

Nowadays, more and more groups aim at directly reprogramming fibroblasts, MG, or other somatic cells into retinal neurons in vivo, and some progress has been made. Mammalian MG cannot be maintained in dishes for a prolonged period, but they can be considered as the endogenous stem cell-like cells, which can be reprogrammed into bipolar, amacrine, and ganglion cells under certain conditions [46,63,81,82,111,112,113]. For example, combining repressing Notch signaling with activating TNFα signaling can stimulate MG proliferation to generate neuronal progenitor cells that subsequently differentiate into retinal neurons [114].

There is no doubt that TFs have played an essential role in the field of cell reprogramming and this will continue. Klf4, which can promote ESC self-renewal [62], is well known as one of the four famous Yamanaka factors [88]. Although Klf4 functions as a transcriptional repressor for axon growth of RGCs and other CNS neurons, it is also a potential candidate factor for reprogramming to replenish RGCs. Rocha-Martins et al. demonstrated that Klf4 was sufficient to change the potency of lineage-restricted retinal progenitor cells to generate RGCs in vivo [115]. 

Another attractive TF, Ascl1, displays a magic ability in cellular reprogramming in the retina. It is capable of reprogramming mouse MG into bipolar and amacrine cells in vitro [116]. The forced expression of this neurogenic TF in MG gave rise to amacrine, bipolar, and photoreceptor cells in young mice after NMDA treatment [113]. In addition, the same group found that MG-specific overexpression of Ascl1, together with NMDA and trichostatin-A, enabled mice to regenerate functional retinal interneurons [46]. Judging from previous data, single Ascl1 is not sufficient to convert MG into RGCs either in vitro or vivo [46,113,116]. Meanwhile, Meng et al. found that adenovirus-mediated transduction of Ascl1, Brn3b(Pou4f2), and Ngn2 can directly convert mouse fibroblasts to RGC-like cells [77]. Recently, Xiao et al. reported that a combination of triple TFs Ascl1, Brn3b/3a, and Isl1 not only reprogrammed fibroblasts into self-organized and networked sensory ganglion organoids but also induced RGCs [80]. Subsequently, Wang et al. confirmed that this cocktail treatment worked well in inducing RGC-like cells [117]. It should be emphasized that strictly speaking, the RGC-like cell is a more proper term for RGCs induced in vitro, because previous studies have shown that it is difficult to distinguish RGCs from peripheral sensory ganglion neurons, since both of them share many common molecular hallmarks. This may no longer be an issue, as our group has recently found that a combination of Pax6 with Brn3a or Brn3b can serve as a unique identifier for RGCs [80].

Apart from Klf4 and Ascl1, Atoh7, an essential basic-helix-loop-helix TF for establishing RGC fate, plays a vital role in RGC regeneration. The forced expression of Atoh7 promotes the differentiation of MG-derived retinal stem cells into RGCs [81,118]. In addition, Dkk1 + Noggin + DAPT and the overexpression of Atoh7 together could convert fibroblasts into RGCs [76]. Furthermore, Neurod1-expressing amacrine and photoreceptor progenitors can be reprogrammed into RGCs when Atoh7 is inserted into the *Neurod1* locus [64]. Xiao et al. showed that combining Atho7 with Brn3b was able to reprogram mature mouse MG into RGCs efficiently and the MG-derived RGCs were functional, made appropriate central projections, and improved visual responses [119]. Ngn2 alone is also sufficient to lineage, reprogramming postnatal mouse MG into RGC-like neurons in vitro and inducing the generation of this neuronal type from late retinal progenitors in vivo [63].

Researchers’ ambitions are unbounded. Numerous modified methods were created to achieve the goal of mature somatic cell-to-neuron conversion. iPSCs were generated from fibroblasts via mRNA reprogramming and subsequently differentiated into a retinal fate by modifying a previously established protocol in a directed, stepwise manner [79]. Most recently, Lu et al. have shown that the ectopic expression of Oct4, Sox2, and Klf4 in RGCs can restore youthful DNA methylation patterns and transcriptomes, promote axon regeneration after injury, and reverse vision loss in models of glaucoma and aged mice [45]. Perhaps, other TFs and epigenetic modifications may also be involved in this process or RGC regeneration. Aside from TF overexpression, knockdown of the RNA-binding protein Ptbp1 by the CRISPR-CasRx system also converted MG into RGCs in mature murine retinas, which alleviates the symptoms associated with RGC loss [82]. Moreover, this approach can also make glia-to-neuron conversion in the brain [82].

## 4. Similarity in Brain Neural Regeneration

The retina, a special part of the CNS, is an excellent and convenient site to explore neuronal regeneration. The strategies of RGC regeneration may be applied to brain neural regeneration and vice versa. Previous studies have shown that transplanting external cells such as neural stem cells, ESC-derived neurons and organoids, or fibroblast-reprogrammed neurons can regenerate neurons in animal brains [97,120,121,122,123,124,125]. Moreover, the in vivo regeneration of GABAergic neurons from astrocytes in Huntington’s disease mouse models can be achieved by AAV-mediated NeuroD1 and Dlx2 delivery [126]. Similarly, in Alzheimer’s disease models, reactive glial cells can be reprogrammed into functional glutamatergic or GABAergic neurons by retroviral expression of NeuroD1 [127]. As mentioned above, in Parkinson’s disease mouse models, direct reprogramming astrocytes into dopaminergic neurons through depleting *Ptbp1* improves Parkinson’s disease-like motor phenotypes [82,128].

Needless to say, local environmental cues are different in different tissues, which may affect the outcome of reprogrammed cells. The methods promoting RGC regeneration in the retina may facilitate the regeneration of other types of neurons in the brain and vice versa. Compared to cell transplantation, in vivo neuronal regeneration by direct reprogramming appears to be a more economical approach, and in general, it makes it easier for the regenerated neurons to migrate to the right place and project to the right targets.

## 5. Conclusions

Several reviews have summarized recent advances in axon regeneration reconnecting the brain [9,129,130,131]. As aforementioned, there are differences in axonal regeneration ability among different RGC subtypes. The presence of numerous intrinsic factors and the extrinsic environment such as TFs, signaling pathways, epigenetic modifications, and inflammatory stimulations suggest that axon regeneration is a complicated process, and no single factor can be adequate to promote regeneration. Additional factors that regulate axon regeneration are likely to be identified, which would benefit us for better understanding the underlying mechanism. Considering that the primary goal of axon regeneration is the restoration of functional visual capacity, there are many questions that remain to be answered. The presence of regenerated RGC axons at one target in the brain does not mean restoring visual function [16,30]. To what extent is axon regeneration is required for restoring meaningful vision? Can it be accomplished by a single or a few factors? How additional manipulations may further improve on currently published approaches? The need to rebuild the precise synaptic connections as in the intact organism and the slow rate of axon regeneration may be a huge hindrance [132]. Experimental axon regeneration is accomplished in rodents or cultured cells. Whether these manipulations are effective for primates and humans remains to be tested.

Based on the published data, Notch, Wnt, BMP, TNFα, insulin-like growth factor signaling pathway, Myt1l, REST, and epigenetic factors all regulate the direct reprogramming procedure [46,61,65,69,74,76,78,111,112,114,133]. In fact, Noggin (BMP signaling pathway inhibitor), Dkk1 (Wnt signaling pathway inhibitor), IGF1, and DAPT (Notch signaling pathway inhibitor) are widely used in many protocols of RGC differentiation from ESCs or iPSCs [61,65,68,69,74,76,78]. We do not know exactly why dozens of treatments involving different signaling pathways can achieve the same reprogramming goal or whether different neuron types can be obtained by combining different epigenetic modification inhibitors. Overall and detailed regulatory networks, without doubt, need to be investigated. As for in vivo RGC regeneration, MG reprogramming has shown great promise. Needless to say, despite the many obstacles and unanswered questions, the outlook of RGC and axon regeneration in vivo is a fascinating and practical possibility.

## Figures and Tables

**Figure 1 ijms-22-04616-f001:**
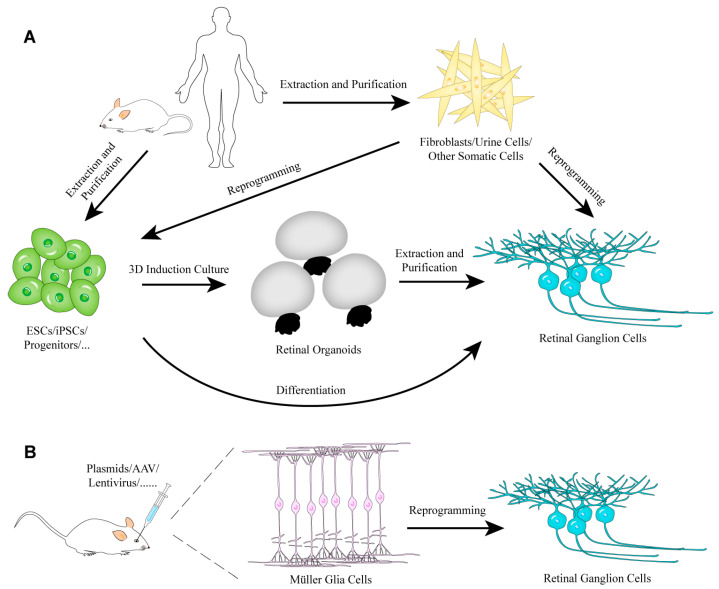
The strategies of RGC regeneration in vitro (**A**) or in vivo (**B**).

**Table 1 ijms-22-04616-t001:** Published retinal ganglion cell induction or regeneration protocols.

Species	Cell Types	Methods	References
*Gallus gallus*	Retinal margin cells	Co-injection of insulin and FGF2	[61]
*Mus musculus*	Retinal progenitor cells	Klf4 overexpression	[62]
*Mus musculus*	Retinal progenitors	Neurog2 overexpression	[63]
*Mus musculus*	Amacrine/photoreceptor progenitors	Replacing Neurod1 with Atoh7	[64]
*Homo sapiens*	ESCs	Noggin + Dkk1 + IGF1	[65]
*Mus musculus*	ESCs	Pax6 overexpression, limiting dilution culture	[66]
*Homo sapiens*	ESCs	N2B27 differentiation media containing 2% Matrigel	[67]
*Homo sapiens*	ESCs	Noggin + Dkk1 + IGF1, extending the duration of embryoid body formation, 10% KSR	[68]
*Homo sapiens*	ESCs	Forskolin + Dorsomorphin + IDE2 + LDN193189 + SB431542 + Nicotinamide + Noggin + DAPT	[69]
*Homo sapiens*	ESCs/iPSCs	Three-dimensional and two-dimensional culture, BDNF supplement	[70]
*Mus musculus*	ESCs/iPSCs	Three-dimensional and two-dimensional culture, BDNF supplement	[71]
*Homo sapiens*	ESCs/iPSCs	20% KSR, FGF2, N2 supplement, B27 supplement	[72]
*Mus musculus*	iPSCs	IMDM, 20% FBS + neural induction, expansion and differentiation medium	[73]
*Mus musculus*	iPSCs	FBS, FGF2, Dkk1, Lefty2, Noggin, N2 supplement, B27 supplement, DAPT, Atoh7	[74]
*Mus musculus*	iPSCs	NeuroD1 overexpression together with retinoid acid and taurine treatment	[75]
*Mus musculus*	Fibroblasts	Reprogrammed into iPSCs, Dkk1 + Noggin + DAPT and Atoh7 overexpression	[76]
*Mus musculus*	Fibroblasts	Ascl1, Brn3b(Pou4f2) and Ngn2	[77]
*Homo sapiens*	Fibroblasts	DKK1 + Lefty A + Noggin, Atoh7 overexpression	[78]
*Homo sapiens*	Fibroblasts	mRNA reprogramming, neural induction and differentiation medium	[79]
*Mus musculus*	Fibroblasts	Ascl1, Brn3b/3a(Pou4f1/Pou4f2) and Isl1	[80]
*Mus musculus*	Müller glia cells	Dedifferentiated into stem cells, then transfected with Atoh7	[81]
*Mus musculus*	Müller glia cells	Neurog2 overexpression	[76]
*Mus musculus*	Müller glia cells	Knockdown of Ptbp1	[82]

## Data Availability

Not applicable.

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
