# Peer review of "Advances in Regeneration of Retinal Ganglion Cells and Optic Nerves"

_ijms, 2021, doi:10.3390/ijms22094616_

Round 1

Reviewer 1 Report

This is a nice review about regeneration in the optic nerve . This is a very complex subject but the authors covered most of the publications in the field. The paper needs a careful editing to improve the clarity.

Author Response

We appreciate the reviewer’s suggestions and comment that “This is a nice review about regeneration in the optic nerve. This is a very complex subject but the authors covered most of the publications in the field.” As suggested, to improve clarity, we carefully read and edited the text and made a number of modifications that are marked in red throughout the revised manuscript.

Reviewer 2 Report

The paper by Yuan, F. is interesting and adds values to the field. There are a couple of points I would like to have to improve the manuscript:

(1) If the authors can introduce a section how the regeneration biology is progressing in case of other organs like brain? This gives an opportunity for the readers to compare this particular field of RGC with other fields.

(2) There are many instances when the authors have used the acronyms of the words like Dkk1, FGF2, Klf4, Atoh7. It will be wonderful if the authors introduce a table with those terms and explain the full forms and functions of those factors/genes/transcripts.

Author Response

We are grateful for the reviewer’s comments and for stating that “The paper by Yuan, F. is interesting and adds values to the field.” The reviewer raised a couple of points that we have addressed below:

(1) Thank you for the comments. We have added a new section (Section 4 marked in red between lines 288 and 305) as suggested to briefly discuss and compare the progress made in brain neural regeneration. We kept the new section short lest it should become a distraction of the main focus on RGC and axon regeneration.

(2) Thank you for the suggestion. As suggested, in the revised manuscript (between lines 343 and 344 and marked in red), we have listed acronyms of all the mentioned factors/genes/transcripts and the corresponding full forms.

Reviewer 3 Report

The authors reviewed advances in the regeneration of retinal ganglion cells and optic nerves. They investigated the factors that affect RGC axon regeneration in a lot of studies. Generation of RGC's from stem cells(ESC's and iPSCs) and generation of RGCs by reprogramming were emphasized by the authors. This review paper is very useful for readers to understand recent advancements in this field of RGC and axon regeneration, differentiation of ESCs and iPSCs into cells with specific RGC characteristics and regeneration reprogramming of RGCs

Author Response

We are grateful for the reviewer’s comments and for stating that “This review paper is very useful for readers to understand recent advancements in this field of RGC and axon regeneration, differentiation of ESCs and iPSCs into cells with specific RGC characteristics and regeneration reprogramming of RGCs.”